# Galangin Rescues Alzheimer’s Amyloid-β Induced Mitophagy and Brain Organoid Growth Impairment

**DOI:** 10.3390/ijms24043398

**Published:** 2023-02-08

**Authors:** Ru Zhang, Juan Lu, Gang Pei, Shichao Huang

**Affiliations:** 1State Key Laboratory of Cell Biology, Shanghai Institute of Biochemistry and Cell Biology, Center for Excellence in Molecular Cell Science, Chinese Academy of Sciences, Shanghai 200031, China; 2Shanghai Key Laboratory of Signaling and Disease Research, Laboratory of Receptor-Based Biomedicine, The Collaborative Innovation Center for Brain Science, School of Life Sciences and Technology, Tongji University, Shanghai 200070, China; 3Institute for Stem Cell and Regeneration, Chinese Academy of Sciences, Beijing 100045, China

**Keywords:** human brain organoid, Alzheimer disease, mitophagy, galangin

## Abstract

Dysfunctional mitochondria and mitophagy are hallmarks of Alzheimer’s disease (AD). It is widely accepted that restoration of mitophagy helps to maintain cellular homeostasis and ameliorates the pathogenesis of AD. It is imperative to create appropriate preclinical models to study the role of mitophagy in AD and to assess potential mitophagy-targeting therapies. Here, by using a novel 3D human brain organoid culturing system, we found that amyloid-β (Aβ_1-42_,10 μM) decreased the growth level of organoids, indicating that the neurogenesis of organoids may be impaired. Moreover, Aβ treatment inhibited neural progenitor cell (NPC) growth and induced mitochondrial dysfunction. Further analysis revealed that mitophagy levels were reduced in the brain organoids and NPCs. Notably, galangin (10 μM) treatment restored mitophagy and organoid growth, which was inhibited by Aβ. The effect of galangin was blocked by the mitophagy inhibitor, suggesting that galangin possibly acted as a mitophagy enhancer to ameliorate Aβ-induced pathology. Together, these results supported the important role of mitophagy in AD pathogenesis and suggested that galangin may be used as a novel mitophagy enhancer to treat AD.

## 1. Introduction

Accumulating evidence suggests that mitochondria, the main energy-producing organelles in cells, play a critical role in the development of Alzheimer’s disease (AD). Mitochondria impairment has been shown to occur in early AD patients, potentially contributing to brain cell death and inflammation, which are hallmarks of Alzheimer’s disease [1]. Mitophagy is the process by which cells remove and recycle impaired mitochondria. This process is important for maintaining the homeostasis of cells, and the dysregulation of mitophagy has been implicated in the progression of neurodegenerative diseases [2,3]. Accumulating evidence indicates that mitophagy is also impaired in AD patient, as well as mouse models, and is linked to the development and progression of the disease in several ways [4,5]. It was reported that intracellular accumulation of tau led to mitophagy deficits and induced neuronal toxicities [6]. Amyloid-β (Aβ), another protein that is thought to play a critical role in the development of AD, was also linked to reduced mitophagy in HEK293 cells and *C. elegans* [5,7]. Moreover, metabolic disorder during AD pathogenesis, for example, the excessive accumulation of cholesterol, was also reported to hamper mitophagic flux and the recycling of impaired mitochondria [8]. Studies have been carried out recently to evaluate the therapeutic effect of mitophagy-inducers on AD. Several naturally occurring compounds, including *P. edulis* extract, kaempferol, and rhapontigenin, have been reported to enhance mitophagy and restore memory deficit in AD *C. elegans* or mice, suggesting that natural products may be valuable in developing novel AD drugs targeting mitophagy [9,10]. Given the critical role of mitophagy in AD pathogenesis, it is important to develop proper models to analyze mitophagy in the context of AD, as well as to develop mitophagy-targeting therapies.

Human brain organoid, also known as a mini brain, or a brain-on-a-chip, is a 3D culture made from human stem cells that is designed to recapitulate the structural and functional features of the human brain [11,12,13,14]. Compared with a 2D cell culture, a brain organoid is composed of various types of brain cells, such as neurons and glial cells [15,16]. These cells grow in a 3D matrix, which allows them to self-organize into a structure that resembles the human brain [17]. Brain organoids are also able to mimic the microenvironment of brain tissues, including nutrients, growth factors, and hormones, as well as the physical and mechanical properties of the surrounding matrix [18]. Based on these features, brain organoids provide a powerful way to study the pathogenesis of many human diseases, including AD [18,19,20,21,22,23]. However, there are few studies exploring the regulatory role of mitophagy in AD using human brain organoids.

In order to develop a brain organoid-based model to study the role of mitophagy during AD pathogenesis, we treated human brain organoids with Aβ and found that organoid growth, as well as mitophagy, was impaired. These results were consistent with observations made in brain samples from AD patients. Using this AD organoid model, we also found that a dietary flavonoid, galangin (gal), restored the impaired mitophagy and organoid growth, indicating its possible functioned as a mitophagy enhancer to ameliorate Aβ-induced pathology. Taken together, this study developed a brain-organoid-based model to study the regulatory role of mitophagy in AD and suggested that galangin may be used as a novel mitophagy enhancer for disease treatment.

## 2. Results

### 2.1. Aβ_1-42_ Suppressed the Growth of Brain Organoids and NPCs, as Well as Impaired Mitochondrial Function

Studies in AD animal models and cell culture experiments strongly indicate that amyloid-β_1-42_ oligomers are the most neurotoxic Aβ species, playing a central role in AD pathogenesis [24]. Thus, we used amyloid-β_1-42_ oligomers in this study. We employed a three-dimensional (3D) brain organoid culture system to investigate the effect of amyloid-β. First, we treated the human brain organoids with Aβ_1-42_ and measured the size of these organoids. Compared with the control, the Aβ_1-42_-treated organoids were smaller in size (Figure 1A,B). We further performed EdU staining in the brain organoids and found that these Aβ_1-42_-treated organoids harbored significantly less EdU^+^ cells compared with the control organoids, indicating that the proliferation of organoids was suppressed by Aβ_1-42_ (Figure 1C,D). Since the growth of brain organoids was mainly contributed by NPCs, we next analyzed the effect of Aβ_1-42_ on NPCs (Appendix A). As shown in Figure 1E, we measured the viability of the NPCs by determining the ATP production using a CellTiter-Glo assay and found that there was a significant decrease in the cell growth after exposure to Aβ_1-42_ for 48 h.

Emerging evidence indicates that mitochondrial dysfunction is a key event in the pathogenesis of AD, and many studies reveal that the mitochondria may be an important target of Aβ [25,26]. We hypothesized that Aβ_1-42_ may alter the growth of NPCs by disrupting mitochondrial function. The levels of reactive oxygen species (ROS) and mitochondrial membrane potential (MMP) were used to assess the function of the mitochondria. At a low membrane potential, the JC-1 dye is present as a monomer (green fluorescence), while at a higher potential, the JC-1 dye monomer forms red fluorescent "J-aggregates" wherever it has accumulated within the mitochondria. Thus, MMP is indicated by the red/green JC-1 fluorescence intensity ratio. As shown in Figure 1F,G, the treatment of Aβ_1-42_ could dramatically increase the level of cellular ROS in the NPCs. Furthermore, we found that Aβ_1-42_ treatment caused a significant loss of MMP, reduced the number of mitochondria, and fragmented the mitochondrial structure in the NPCs (Figure 1H,I and Appendix A), suggesting that mitochondria function was impaired in the Aβ_1-42_ group. 

### 2.2. Mitophagy Was Impaired by Aβ_1-42_ in Brain Organoid and NPCs

Since we observed that Aβ_1-42_ impaired the mitochondrial function of NPCs, we hypothesized that mitophagy, which is essential for the recycling of mitochondria, may be affected. To investigate the impact of Aβ_1-42_ on mitophagy, we performed mitophagy staining in organoids and found that it was significantly suppressed by Aβ_1-42_ (Figure 2A,B). We also analyzed the level of mitophagy-related protein PINK1 by Western blotting. The result showed that the PINK1 protein level in the Aβ_1-42_ treated organoids was decreased (Figure 2C,D). In accordance with the results in organoids, the mitophagy in NPCs was significantly decreased after Aβ_1-42_ treatment (Figure 2E,F and Appendix A). The protein levels of PINK1, LC3 I/II, and Beclin-1 decreased in a dose-dependent manner after Aβ_1-42_ treatment in NPCs, while the protein level of the autophagy substrate p62 increased significantly. Meanwhile, the control reverse peptide Aβ_42-1_ had no effect on the expression level of these proteins (Figure 2G–K). These results indicated that Aβ_1-42_ markedly impaired PINK1-mediated mitophagy in organoids and NPCs.

### 2.3. Galangin Rescued Aβ-Induced Mitophagy Impairment in Brain Organoids

Previous studies reported that flavonoid is associated with lower risk of AD [27]. Here, we focused on the effects of galangin, which is a low toxicity, naturally occurring dietary flavonoid [28]. It has been reported that galangin affects several aspects in AD pathologies, such as reducing p-tau, Aβ_1-42_, and β-secretase [29,30,31]. To investigate whether galangin ameliorates the Aβ_1-42_-induced mitophagy impairment, we cultured Aβ_1-42_-treated organoids, with or without galangin. As shown in Figure 3A,B, 10 μM galangin significantly increased mitophagy intensity in the Aβ_1-42_+gal group compared to the Aβ group. Additionally, the protein level of PINK1 increased in the Aβ_1-42_+gal group compared to the Aβ_1-42_ group (Figure 3C,D). These results suggested that galangin ameliorated the Aβ-induced mitophagy impairment in brain organoids.

Furthermore, we examined the effect of galangin on the growth of Aβ_1-42_-treated organoids. As shown in Figure 3E,F, the Aβ_1-42_+gal group was increased in size compared to the Aβ_1-42_ group. Similarly, we found that the protein level of PINK1 was also significantly reduced by Aβ_1-42_ treatment in NPCs, which was elevated by galangin (Figure 3G,H). These results suggest that galangin may have a protective effect against the impairment of mitophagy and the inhibition of organoid growth caused by Aβ_1-42_.

### 2.4. Inhibition of Mitophagy Abolished the Effects of Galangin

Given the above results, we hypothesized that galangin promoted the growth of brain organoids and NPCs through enhancing mitophagy. To investigate this, we treated the Aβ_1-42_+gal group with Mdivi-1, a mitophagy inhibitor, and assessed its impact on the effects induced by galangin. First, the inhibitory effect of Mdivi-1 on the mitophagy of organoids was verified by mitophagy staining and Western blotting (Figure 4A–C). We then analyzed the effects of Mdivi-1 on the growth of brain organoids. The results showed that Mdivi-1 treatment abolished the rescue effect of galangin on organoid size (Figure 4D,E). Similarly, we found that galangin treatment alleviated the Aβ_1-42_- induced suppression of NPC growth, which was also blocked by Mdivi-1 (Figure 4F). Together, these results indicated that galangin treatment rescued the Aβ_1-42_-induced impairment of organoid and NPC growth through enhancing mitophagy.

## 3. Discussion

AD is tightly linked to metabolic changes in the brain, including alteration in glucose and fatty acid metabolism, mitochondrial function, and mitophagy, as well as inflammation. Considering the energy coupling between different brain cell types, it is reasonable that different cell types are involved in the regulation of these changes. Thus, to study the overall impact of different cells and microenvironment on mitophagy in the context of AD, we treated brain organoids with Aβ to mimic the real situation occurring in the brains of AD patients. Using WB and mitophagy dye staining assays, we found that mitophagy was significantly reduced in brain organoid, which was consistent with the previous findings from AD brains, suggesting that the brain organoid culturing system was a reliable method for modeling the changes in mitophagy that occur in AD. Moreover, our results showed that mitophagy impairment was associated with the reduction in organoid and NPC growth, suggesting that the dysregulation of mitophagy in the brain should also be corrected in addition to regenerative therapies for AD to improve their efficiency.

Galangin is a low toxic, naturally occurring dietary flavonoid compound that is found in honey and ginger plants. It has been studied for its potential medicinal properties, including anti-inflammatory and anticancer effects. We first demonstrated that galangin treatment rescued the Aβ-induced impairment of mitophagy, proposing that galangin, alone or in combination with other mitophagy enhancers such as urolithin A, NAD^+^ boosters, and metformin, might potentially be used as a safe and effective treatment for diseases associated with impaired mitophagy. In addition to its ability to enhance mitophagy, galangin has been reported to reduce excessive inflammation, which is also a hallmark of AD [32]. These findings suggests that galangin may serve as a multi-targeted drug for the treatment of AD.

Defective mitophagy is associated with other key AD pathologies, including Aβ and tau [5]. Interestingly, galangin is reported to regulate the β-Secretase BACE1, which is critical for Aβ production, by inducing a decrease in acetylated H3 in the BACE1 promoter regions [31]. Our findings, along with those in this study, raise the possibility that mitophagy may modulate Aβ production through epigenetic regulation. Moreover, this hypothesis may prove to be true for other mitophagy inducers. Deciphering the complicated interaction between mitophagy and other AD pathologies through mechanistic studies on these mitophagy inducers may possibly provide novel innervations for AD.

In summary, we demonstrated that mitophagy was significantly reduced by Aβ treatment, suppressing the growth of brain organoids and NPCs. The dietary flavonoid galangin restored the mitophagy impairment by Aβ, and its rescue effect could be blocked by the mitophagy inhibitor. Further in vivo experiments in AD animal models are required to test the effect of galangin on the improvement of cognitive function. Our findings suggest that galangin could potentially serve as a novel treatment for AD.

## 4. Materials and Methods

### 4.1. Cell Culture

Human iPSC/iPSC-derived NPCs were provided by IxCell Biotechnology Ltd. The human iPSCs-derived NPCs cells were maintained as an adherent culture in 50% DMEM-F12 and 50% Neurobasal Medium, containing N2 supplement, B27 supplement (Minus Vitamin A), NEAA, 1 × Glutamax, 10 ng/mL FGF-Basic Recombinant Human Protein (bFGF, Gibco, New York, NY, USA), 10 ng/mL LIF Recombinant Human Protein (hlif, Gibco, New York, NY, USA), 5 μM SB431542 (Selleckchem, Shanghai, China), 3 μM CHIR99021 (Selleckchem, Shanghai, China), and 200 μM L-Ascorbic acid 2-phosphatesesquimagnesium salt hydrate (Sigma, St. Louis, MO, USA). The cells were maintained at 37 °C in humidified air with 5% CO_2_.

### 4.2. Compounds and Reagents

Galangin was purchased from Biopurify Phytochemicals Ltd (Chengdu, China), and DAPI and Mdivi-1 were purchased form Beyotime (Shanghai, China). CellTiter-Glo was purchased from Vazyme Biotech Co.,Ltd (Nanjing, China). Mitophagy dye was obtained from DOJINDO (Kumamoto, Japan).

### 4.3. Aβ_1-42_ Oligomers Preparation

Aβ_1-42_ peptides were purchased from Chinese Peptide Company (AMYD-003). A total of 2 mg Aβ_1-42_ peptide was dissolved in 2 mL of hexafluoroisopropanol (HFIP) (Sigma, St. Louis, MO, USA), and was then dispensed into Protein LoBind tubes (Eppendorf, 030108094) and dried overnight at room temperature (RT) to form a clear dried film. HFIP-treated Aβ_1-42_ peptides were resuspended in dimethyl sulfoxide (DMSO) and then diluted to 100 μM with phenol-red free DMEM/F12 medium. The diluted Aβ_1-42_ peptides were then vortexed for 30 s, followed by incubation for 24 h at 4 °C to form oligomers.

### 4.4. Human Brain Organoid Culture

On day 0, human NPCs at 90% confluence were dissociated into single cells using Accutase (5 min, 37 °C). After centrifugation at 1000 rpm for 5 min, the NPCs were resuspended with NPC medium. Each brain organoid was generated from 10,000 cells and cultured in an ultra-low-attachment 96-well U-bottom plate (Corning, New York, NY, USA) in the presence of ROCK inhibitor Y-27632 (10 μM). After centrifugation at 100× *g* for 5 min, the plates were maintained at 37 °C in humidified air with 5% CO_2_. The medium was replenished every 2 days. On day 4, the spheres were transferred to an ultra-low-attachment 48-well plate (Corning) and were kept on an orbital shaker at a speed of 80 rpm/min. The medium was replenished every 3 days. On day 20, the organoids were transferred to a differentiation medium, which is comprised of a 1:1 mixture of Neurobasal and DMEM/F12, supplemented with N2 (Thermo Fisher Scientific, New York, NY, USA), BDNF (20 ng/mL, Pepro Tech, NJ, USA), GDNF (20 ng/mL, Pepro Tech, NJ, USA), dibutyryl-cyclic AMP (200 μM, MCE, Shanghai, China), and ascorbic acid (200 μM, Sigma, St. Louis, MO, USA). Aβ_1-42_ and chemicals were added when organoids were transferred to the differentiation medium. Every other day, images of the organoids were captured the size of the organoids was analyzed with Image J software. On day 28, the organoids were harvested for testing.

### 4.5. EdU Detection

For EdU detection, EdU (5 μM, Sigma, St. Louis, MO, USA) was added and incubation was conducted for 0.5 h before harvesting the organoids. The organoids were fixed with 4% PFA for 1 h and allowed to precipitate in 30% sucrose solution for 48 h. The organoids were cryosectioned at 20 µm and incubated with PBS containing 0.3% Triton X-100 at room temperature (RT) for 15 min; these samples were then incubated with Click reaction solution at RT for 30 min, protected from light. After three washes with PBS, these samples were stained with DAPI. The numbers of EdU-positive cells were analyzed by ImageJ.

### 4.6. Cell Viability Assay

Cell viability was measured by an ATP assay using a CellTiter-Glo Luminescent kit, according to the manufacturer’s instructions. On day 0, NPCs were plated in Matrigel-coated 96-well plates at 2000 cells per well in medium consisting of 1:1 DMEM/F12: Neurobasal, NEAA, Glutamax, N2 and B27 supplement, 1 ng/mL bFGF. On day 1, the NPCs were treated with Aβ_1-42_ or other chemicals. ATP assay was performed on day 3. Luminescence was detected with a BioTek SynergyNEO microplate reader (Bio-Tek, Williston, VT, USA).

### 4.7. ROS Assay

DCFH-DA (#S0033, Beyotime, Shanghai, China) was used to detect intracellular ROS levels. Human NPCs were plated in Matrigel-coated 96-well black plates (#3094,Corning, NY, USA) at 2000 cells per well and cultured in medium consisting of 1:1 DMEM/F12: Neurobasal, NEAA, Glutamax, N2 and B27 supplement, 1 ng/mL bFGF. The cells were treated with 10 μM Aβ_1-42_ for 24 h and then stained with 10 μM DCFH-DA and 3 μg/mL Hoechst (#C1022, Beyotime, Shanghai, China) for 30 min at 37 °C in a humidified incubator with 5% CO_2_. The cells were washed twice with PBS and observed under a laser-scanning confocal microscope (Operetta, Perkin Eimer, MA, USA). Alternatively, the fluorescence was measured by a BioTek SynergyNEO microplate reader (Bio-Tek, Williston, VT, USA) at a 488 nm excitation wavelength and a 525 nm emission wavelength.

### 4.8. Measurement of Mitochondrial Membrane Potential

JC-1 staining kits (Beyotime, C2006) were used to assess the MMP level of the cells, according to the manufacture’s protocols. Briefly, NPCs were plated in Matrigel-coated 96-well black plates (#3094,Corning, NY, USA) at 2000 cells per well and cultured in medium consisting of 1:1 DMEM/F12: Neurobasal, NEAA, Glutamax, N2 and B27 supplement, 1 ng/mL bFGF. After 24 h of treatment with 10 μM Aβ_1-42_, 50 μl of JC-1 staining solution was added to each well and incubated at 37 °C for 20 min. These cells were then washed twice with the staining buffer and observed under a Zeiss Observer Z1 microscope. The fluorescence intensity was detected using the BioTek SynergyNEO (BioTek, Williston, VT, USA) device. The JC-1 monomer was detected at a 490 nm excitation wavelength and a 530 nm emission wavelength. The JC-1 polymers (J-aggregates) were detected at a 525 nm excitation wavelength and a 590 nm emission wavelength. The membrane potential was represented as the ratio of red/green fluorescence intensity.

### 4.9. Mitophagy Detection

Human brain organoids were treated with Aβ_1-42_ or other chemicals for 8 days, then 0.1 μmol/L Mtphagy dye (DOJINDO) and 3 μg/mL Hoechst were added to human brain organoids and incubated at 37 °C for 30 min. These organoids were then washed with PBS, embedded in 50% Matrigel to avoid unwanted movement, and observed under a confocal microscope (FV10-ASW 4.2; Olympus, Tokyo, Japan). As for the NPCs, after 48 h of Aβ_1-42_ treatment, mitophagy detection was conducted using the same method as that used for the organoids.

### 4.10. Mitochondrial Content and Morphology Analysis

Mitochondrial morphology was analyzed by staining with MitoTracker Green (Beyotime), following the manufacturer’s protocol. NPCs were incubated with MitoTracker Green (50 nM) at 37 °C for 45 min. Then, the cells were imaged with an Olympus FV1200 confocal microscope. ImageJ, with a macro developed by Ruben K. Dagda, was used to quantify the mitochondrial morphology [33]. Briefly, the area of interest (one cell) was selected with the polygon selection tool. Then, the image was sharpened and automatically thresholded. Next, the mitochondrial content and circularity were obtained using the analyze particles tool.

### 4.11. Western Blot

Cells (2 × 10^5^ cells/well) were treated with Aβ_1-42_ or other chemicals for 48 h and lysed with 1× Laemmli buffer. The samples were separated by sodium dodecyl sulfate-polyacrylamide gel electrophoresis (SDS-PAGE) and transferred onto nitrocellulose membranes (400 mA constant current, 2 h, 4 °C). The membranes were blocked with 5% nonfat milk in TBS containing 0.1% Tween-20 (TBST) for 0.5 h at RT. The membranes were subsequently incubated with PINK1 (#6946, 1:1000, CST, Danvers, MA, USA) and actin (#A2066, 1:1000, Sigma, St. Louis, MO, USA) antibodies at 4 °C overnight. After three washes with TBST, the membranes were incubated with horseradish peroxidase- (HRP) conjugated secondary antibody for 1 h at RT. After three washes with TBST, the membranes were then incubated with ECL substrate and visualized using the Mini Chemiluminescent Imaging and Analysis System.

### 4.12. Immuostaining

NPCs were fixed in 4% PFA for 15 min and permeabilized with 0.1% Triton X-100 for 30 min. The cells were then incubated with rabbit anti-Nestin antibody (1:100; Abclonal, Wuhan, China) and mouse anti-Aβ antibody (1:100, Covance, NJ, USA) at 4 °C overnight. After three washes with PBS, the cells were incubated with donkey anti-rabbit Alexa 488 (1:1000, Molecular Probes, New York, NY, USA) and donkey anti-mouse Cy3 (1:1000, Molecular Probes, New York, NY, USA) for 1h at RT. All images were analyzed with an Olympus FV10i confocal microscope.

### 4.13. Statistical Analysis

Data are presented as mean ± SEM. Statistical analyses were performed using Prism 6.0 (GraphPad Software Inc., San Diego, CA, USA). An unpaired Student’s *t*-test (two-tailed) was applied for the comparisons of two datasets, and one-way or two-way analysis of variance (ANOVA) with Bonferroni’s post hoc test was used, when more than two datasets were compared. Statistical significance was accepted at *p* < 0.05.

## Figures and Tables

**Figure 1 ijms-24-03398-f001:**
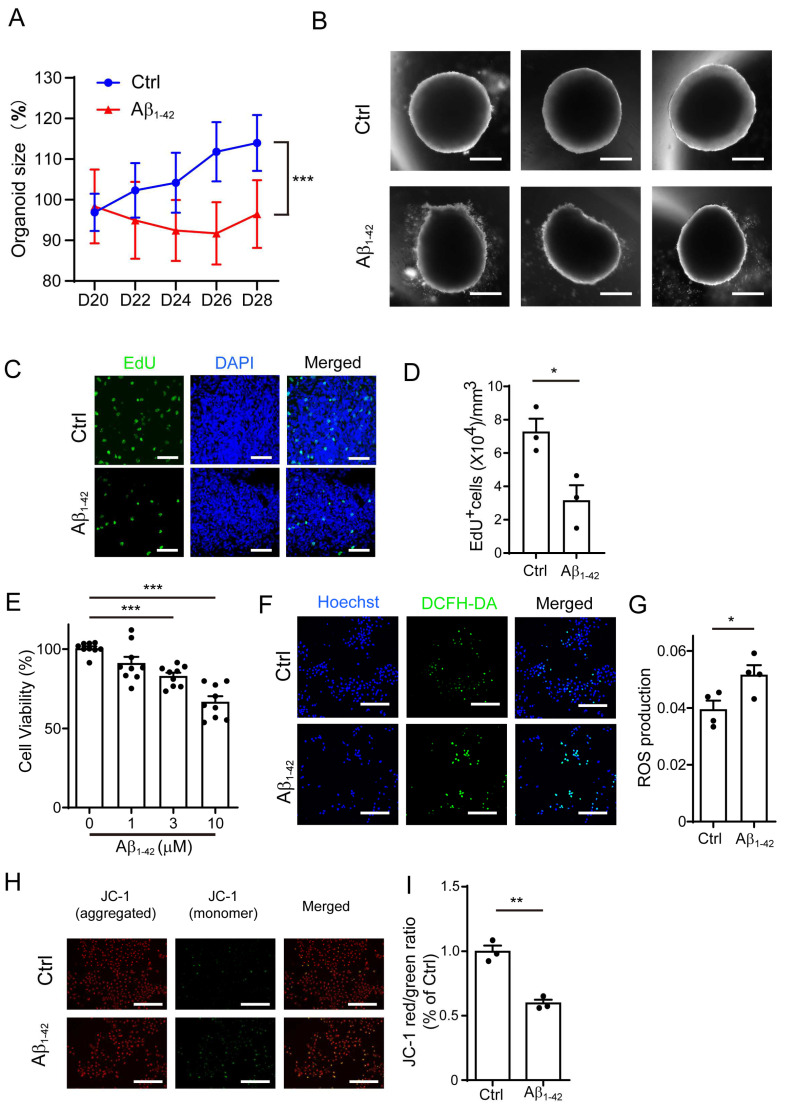
Aβ_1-42_ suppressed the growth of brain organoids and NPCs and impaired mitochondrial function. (**A**) The growth curve of brain organoid with/without Aβ_1-42_ (10 μM) treatment for 8 days (D20-28). (**B**) The representative images from (**A**); scale bars, 300 μm. (**C**) EdU staining of brain organoid with/without Aβ_1-42_ (10 μM) treatment; scale bars, 50 μm. (**D**) The quantification of (**C**). (**E**) NPC cell viability treated with different concentrations of Aβ_1-42_ for 48 h. (**F**) The representative ROS images in NPCs incubated with Aβ_1-42_ (10 μM) for 24 h; scale bars, 200 μm. (**G**) The quantification of (**F**). (**H**) The representative images of JC-1 staning in NPCs incubated with Aβ_1-42_ (10 μM) for 24 h; green—excitation: 490; emission: 530; red—excitation: 525; emission: 590; scale bars, 200 μm. (**I**) The ratio of red/green fluorescence from (**H**). The data are presented as mean ±SEM, *n* ≥ 3 independent experiments; * *p* < 0.05, ** *p* <0.01, *** *p* < 0.001, analyzed by one-way ANOVA followed by Bonferroni’s test.

**Figure 2 ijms-24-03398-f002:**
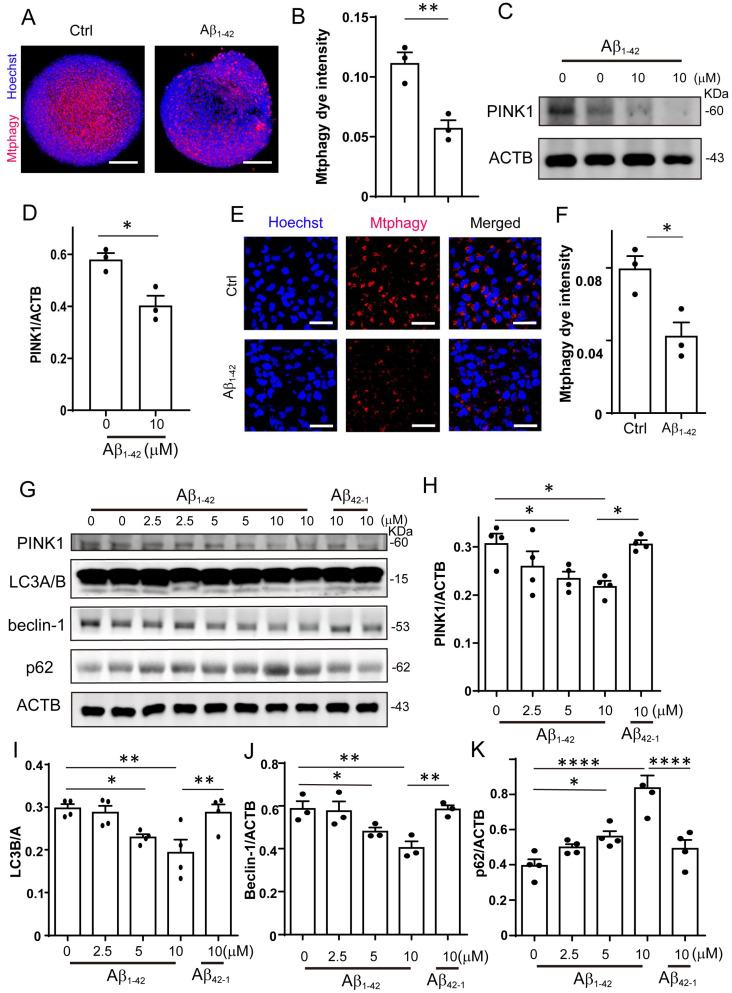
Mitophagy was impaired by Aβ_1-42_ in brain organoid and NPCs. (**A**) The representative images of mitophagy in organoids treated with Aβ_1-42_ (10 μM) for 8 days; scale bars, 200 μm. (**B**) The quantification of mitophagy intensity of (**A**). (**C**) The protein level of PINK1 was determined by Western blotting under Aβ_1-42_ (10 μM) treatment in organoids. (**D**) The quantification of the relative expression level of PINK1 in (**C**). (**E**) The representative images of mitophagy in NPCs treated with Aβ_1-42_ (10 μM) for 48 h; scale bars, 50 μm. (**F**) The quantification of mitophagy intensity of (**E**). (**G**) The protein level of PINK1, LC3A/B, Beclin-1, and p62 were determined by Western blotting under Aβ_1-42_ (10 μM) treatmen in NPCs. (**H**–**K**) The quantification of the relative expression level of PINK1, LC3A/B, Beclin-1, and p62 in (**E**). The data are presented as mean ± SEM, *n* ≥ 3 independent experiments; * *p* < 0.05, ** *p* < 0.01, **** *p* < 0.0001, analyzed by one-way ANOVA followed by Bonferroni’s test.

**Figure 3 ijms-24-03398-f003:**
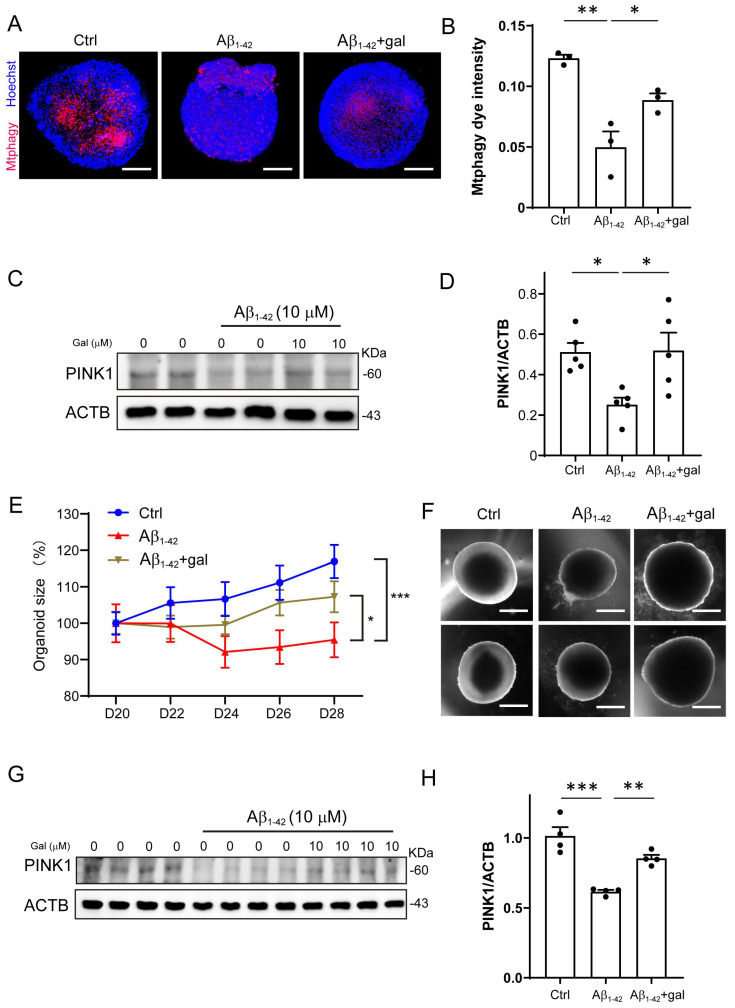
Galangin rescued Aβ-induced mitophagy impairment in brain organoids. (**A**) The representative images of mitophagy in organoids treated with Aβ_1-42_ (10 μM) and galangin for 8 days; scale bars, 200 μm. (**B**) The quantification of (**A**). (**C**) The protein level of PINK1 was determined by Western blotting under Aβ_1-42_ (10 μM) and galangin (10 μM) treatment in organoids. (**D**) The quantification of (C). (**E**) The growth curve of cerebral organoid with Aβ_1-42_ and galangin treatment for 8 days (D20-28). (**F**) The representative images of brain organoid with Aβ_1-42_ and galangin treatment; scale bars, 300 μm. (**G**) The protein level of PINK1 was determined by Western blotting under Aβ_1-42_ (10 μM) and galangin treatment in NPCs. (**H**) The quantification of (**G**). The data are presented as mean ± SEM, *n* ≥ 3 independent experiments; * *p* < 0.05, ** *p* < 0.01, *** *p* < 0.001, analyzed by one-way ANOVA followed by Bonferroni’s test.

**Figure 4 ijms-24-03398-f004:**
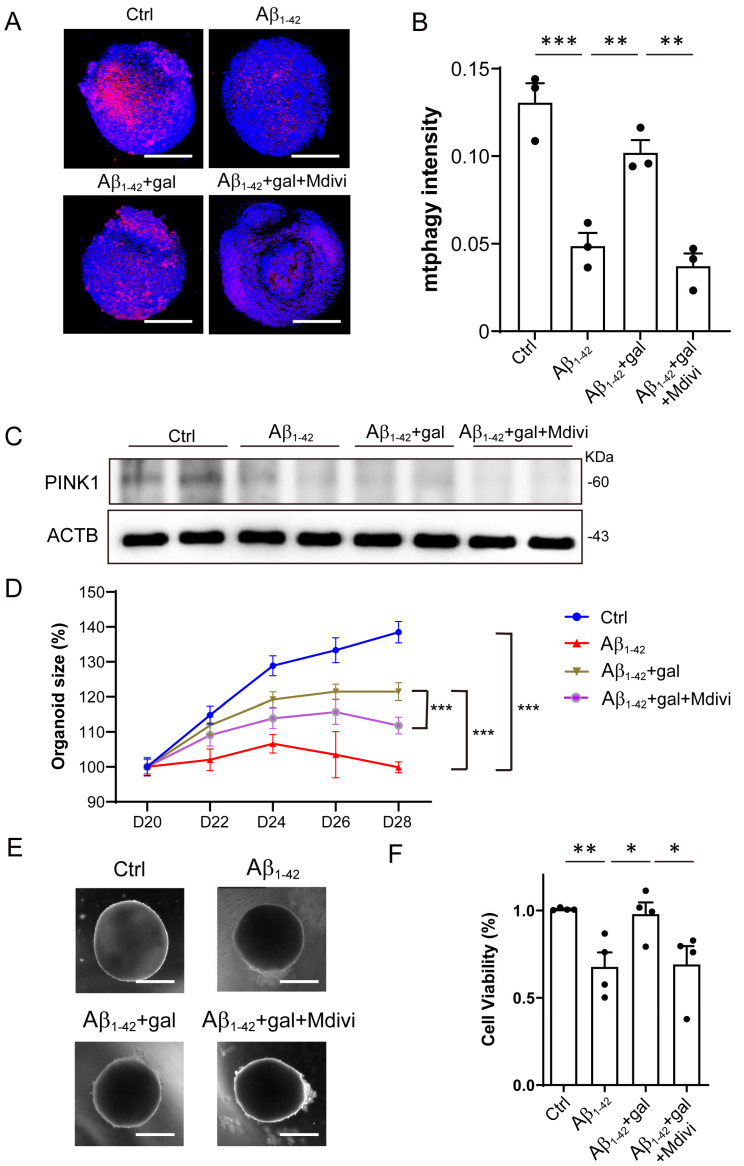
Inhibition of mitophagy abolished the effects of galangin. (**A**) The representative images of mitophagy in organoids treated with/without Aβ_1-42_ (10 μM), galangin (10 μM), and Mdivi-1 (10 μM) for 8 days; scale bars, 200 μm. (**B**) The quantification of (A). (**C**) The protein level of PINK1 was determined by Western blotting, with/without Aβ_1-42_ (10 μM), galangin (10 μM), and Mdivi-1 (10 μM). (**D**) The growth curve of cerebral organoid, with/without Aβ_1-42_, galangin, and Mdivi-1 treatment for 8 days (D20–28). (**E**) The representative images of brain organoid treated with/without Aβ_1-42_, galangin, and Mdivi-1 treatment; scale bars, 300 μm. (**F**) NPCs cell viability treated with/without Aβ_1-42_, galangin, and Mdivi-1 for 48 h. The data are presented as mean ± SEM, *n* ≥ 3 independent experiments; * *p* < 0.05, ** *p* < 0.01, and *** *p* < 0.001, analyzed by one-way ANOVA followed by Bonferroni’s test.

## Data Availability

The datasets in this study are available from the corresponding author upon reasonable request.

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
