# Peer review of "Galangin Rescues Alzheimer’s Amyloid-β Induced Mitophagy and Brain Organoid Growth Impairment"

_ijms, 2023, doi:10.3390/ijms24043398_

Round 1
Reviewer 1 Report
The authors have designed and performed the experiment to study that galangin can rescue the cells by enhancing mitophagy. Overall, the study is complete and carefully performed. The manuscript can be accepted after minor revision.
Minor points:
1. In the main text, the author should mention what kind of amyloid species they used in the experiment. They should add a few words and references that they used amyloid oligomers, which are the most neurotoxic amyloid species, in the studies.
2. A few studies have already demonstrated that galangin can decrease the level of amyloid species and other hallmarks in AD. The authors should add more references in the manuscript.
3. There are also some other proteins involved in the process of mitophagy, for example, LC3. Have the authors checked the level of other proteins?
Reviewer 2 Report
In the paper entitled «Alzheimer’s amyloid-beta impairs mitophagy and growth of brain organoids which could be rescued by dietary flavoid galangin», the authors use Abeta1-42 treatment in 3D organoids and NPCs as a model of Alzheimer’s disease.
They further treat the two cell-based systems with a flavonoid Galangin (Gal). The authors write that mitochondria and mitophagy is impaired by treatment with Abeta and that Gal can restore mitophagy.
The world is in need of novel compounds and anti-AD treatments, due to the increasing incidences of AD and other neurodegenerative diseases. However, this reviewer believes that the concluding points in the paper is overselling the actual data presented. Mitophagy cannot be measured only by looking at PINK1 levels (which has not even been shown the size off and therefore it is not clear whether it is full length PINK1) together with a mitophagy dye not specified. And despite the promising previous published data on brain organoids, it cannot tell us the function of the drug in a system, the availability to cross a brain-barrier function, the ability to reduced memory/cognition etc. The reviewer understand that it might not be possible with cross species approached, but the limitations of ones study should be mentioned.
There are therefore multiple experiments lacking before the conclusions are supported by experimental results. Furthermore, the language needs to be improved. Some places, the meaning of the sentences are unclear, like when writing in consistent – does it mean inconsistent or in consistency?
The following comments need to be answered before publication can be considered.
Title: the title is too long and should be sharpened.
In the introduction, references on mitophagy-inducers and Alzheimer’s disease are lacking, including a recent published paper on extracts from P. edulis, fitting nicely with the story presented here (34992270, 35739408).
It would help the reader, if the authors included an explanation including references on why Gal was used in the paper? Also, are any known side effects published before, any other papers looking at Gal?
In Fig. 1, the authors show that treatment with Abeta1-42 reduces size and proliferation of brain organoids and NPCs.
1) The authors should show whether Abeta is taken up by the organoids and NPCs by stainings. Furthermore, it would be interesting to stain for different cell types in the organoids, and show which cell types are correlating with Abeta staining.
2) In line 79, the authors should explain Why and add references.
3) In line 80, the authors should explain how JC-1 can be used to measure mitochondrial membrane potential with a few words.
4) It is interesting to measure ROS as a measure of cellular stress level or indications of mitochondrial function. However, the authors should write whether it is cellular or mitochondrial ROS measured? It says in the methods, but should also be clear in the text.
5) It would support the conclusion that mitochondria are dysfunctional and mitophagy is compromised, by staining the organoids and / or NPCs with a mitochondrial marker to show both the amount of mitochondria, and the network/structure +/- Abeta treatment.
6) It should also be mentioned how cell viability was measured (Fig. 1E).
When explaining fig. 3, references are lacking on line 120-121 on Gal toxicity.
In fig. 4C, the actin image looks wrong. Please, make sure that it has been inserted correct and in its original size.
In Fig. 2, the authors show that mitophagy is impaired by Abeta treatment, and in Fig. 3-4 that Gal works through mitophagy activation.
In general, the reviewer does not think it is sufficient to only use a mitophagy dye and PINK1 protein level (full length?) to state whether mitophagy is impaired. Therefore, more experiments are needed throughout the manuscript to ensure that this statement/conclusion is correct. Therefore the following comments are for Fig. 2-4.
1) For all WBs molecular sizes are lacking, therefore they cannot be taking into account in their current state.
2) No images of the NPCs are presented, therefore it is not clear how mitophagy is changing here, only that PINK1 (might) change.
3) An array of mitophagy related proteins (and their activation status, phosphorylation) should be accessed with WB or other methods in addition to the mitophagy dye already used. Also, it should be clear which dye was used in the paper. Further, controls on mitophagy induction is not present in any experiment.
4) Dose-dependent experiments or optimization of the dose of Gal should be done, to ensure that the dose used is the optimal dose for mitophagy induction. If done in previous studies, it should at least be referred to. The same goes for the mdivi-1 mitophagy inhibitor experiments.
5) It would have helped to support the important role of PINK1 in the function of Gal, if the authors had make PINK1 KD experiments in the NPCs and showed the effect of Gal in addition to the mdivi-1 experiments.
Reviewer 3 Report
This paper describes the role of Ab amyloid in the growth and development of brain organoids and mitochondrial functions, especially mitophagy. While the manuscript and the research idea is good, there needs to be significant improvement in the paper. Below are my comments for this paper.
1) Fig 1B. The authors have shown images of healthy and Ab treated brain organoids. While looking at them, it seems that the Ab brain organoids are smaller, but the authors need to use softwares such as Image J to determine the size of the brain organoids and then plot the area to give a more accurate representation.
2) Fig 1A. From this graph it seems that organoid size was looked at D8. Is there a rational to show brain organoid size at D8 when they were harvested at D28? What does the organoid size look at D28?
3) In the field of organoid research, organoids are mostly harvested at D60 or later. This is because there is enough neuronal maturation to investigate different parameters. At D28, the neurons are just beginning to mature which requires the mitochondria. Why were the organoids not harvested at a later stage.
4) With most JC1 dyes, the aggregates are red and the monomers are green. It seems like the authors have them reversed. Moreover there does not seem to be to much difference between the healthy and Ab treated.
5) The material and methods section is not well written for mitophagy detection. There is no mention of what the mitophagy dye is and there is also no mention of the methods for staining the brain organoids with the dye. Were the brain organoids sectioned? if so what thickness? Be thorough in the methods.
6) Fig 2C. Is there a rationale why the concentration experiments for Ab treatment was used in duplicates?
7) Fig 2E. There does not seem to be a significant difference between the bands for the Ab treatment and the reverse.
8) Fig 4C. The ACTB bands are not of the same intensity and therefore it'll be very hard to make any comparison between treated and untreated groups.
9) Fig 4E. Could you calculate area for these organoids to show difference in organoid growth?
10) Discussion: The discussion for this manuscript is not well written and just talks about the result. Discussion should be more about perspective and therefore this needs to be re-written.
Round 2
Reviewer 2 Report
After the revision of the manuscript, the experimental results support the conclusions drawn by the authors.
The reviewer is therefore satisfied with the revision and the response to the comments given. Also in light of the short timeframe for revision (10 days), it is satisfactory. It would still be interesting to validate the proposed mechanism by doing knock down of the line of mitophagy-related genes/proteins suggested to be important for the role of Gal.
The reviewer suggest that the English in the manuscript should be improved/checked before publication to ensure clarity.
Author Response
We thank the reviewer's advice to improve the manuscript. According to your suggestion, we have checked and revised our manuscripts. And we agree that mitophagy-related genes/proteins knockdown experiments will certainly help to support the important role of gal, we will perform this experiment in the future.
Reviewer 3 Report
Thank you for the authors for their response. I believe the authors have adequately responded to my comments.
Author Response
Thank you for your careful review. According to your suggestion, we have revised our manuscript again.